# Vapour-phase-transport rearrangement technique for the synthesis of new zeolites

Valeryia Kasneryk[1,5], Mariya Shamzhy[1,5], Jingtian Zhou[2,5], Qiudi Yue [1], Michal Mazur[1], Alvaro Mayoral [3], Zhenlin Luo[2]*, Russell E. Morris [1,4], Jiří Čejka[1] & Maksym Opanasenko[1]*

Owing to the significant difference in the numbers of simulated and experimentally feasible zeolite structures, several alternative strategies have been developed for zeolite synthesis. Despite their rationality and originality, most of these techniques are based on trial-and-error, which makes it difficult to predict the structure of new materials. Assembly-Disassembly-Organization-Reassembly (ADOR) method overcoming this limitation was successfully applied to a limited number of structures with relatively stable crystalline layers (**UTL**, **UOV**, ***CTH**). Here, we report a straightforward, vapour-phase-transport strategy for the trans-formation of **IWW** zeolite with low-density silica layers connected by labile Ge-rich units into material with new topology. In situ XRD and XANES studies on the mechanism of **IWW** rearrangement reveal an unusual structural distortion-reconstruction of the framework throughout the process. Therefore, our findings provide a step forward towards engineering nanoporous materials and increasing the number of zeolites available for future applications.

[1] Department of Physical and Macromolecular Chemistry, Faculty of Science, Charles University, Hlavova 8, 128 43, Prague, Czechia. [2] National Synchrotron Radiation Laboratory, University of Science and Technology of China, 230026 Hefei, Anhui, P. R. China. [3] Center for High-resolution Electron Microscopy (ChEM), School of Physical Science and Technology ShanghaiTech University, 393 Middle Huaxia Road, 201210 Pudong, Shanghai, China. [4] EaStCHEM School of Chemistry, University of St Andrews, Purdie Building, St Andrews KY16 9ST, UK. [5]These authors contributed equally: Valeryia Kasneryk, Mariya Shamzhy, Jingtian Zhou. *email: zlluo@ustc.edu.cn; maksym.opanasenko@natur.cuni.cz

Zeolites are crystalline porous materials used for many applications, including gas separation and catalysis[1–3]. Although millions of thermodynamically stable structures have been predicted under ambient conditions, until now, the zeolite community has only recognised ~250 different zeolite topologies[4]. Such discrepancy between the numbers of proposed zeolite topologies and those produced via traditional hydrothermal approaches has prompted the development of alternative strategies for zeolite synthesis. These new strategies include both direct (e.g., using phosphorus-containing cations[5,6], metal complexes[7] or proton sponges[8] as structure-directing agents) and post-synthesis (3D-3D transformation at high pressures >1 GPa[9]; 3D-2D-3D transformation known as ADOR[10]) methods. However, most of these approaches use a trial-and-error tactic, except for ADOR[11]. ADOR is a unique approach because the topology of new zeolites can be easily predicted based on the knowledge of the parent structure. ADOR strategy was successfully applied for the structural reconstruction of several germanosilicate zeolites such as **UTL**[10], **UOV**[12] and **\*CTH**[13] (three-letters codes are assigned to established structures of zeolites that satisfy the rules of the IZA Structure Commission). However, attempts to transform other potential structures (e.g., **IWW**, **ITH** and **ITR**, among others) using this approach have been unsuccessful thus far because "open framework" zeolite layers containing pores perpendicular to the layer plane are highly labile, resulting in the complete degradation of the zeolite layers or, at best, in the reconstruction of the initial zeolite framework with a chemical composition different from that of the parent material[14].

Here, we report a straightforward strategy to construct new zeolites through non-contact vapour-phase-transport (VPT) rearrangement at room temperature. This method offers the opportunity to prepare new zeolite topologies otherwise inaccessible by both hydrothermal and conventional ADOR synthesis routes. We have used a combination of in situ (XRD, XANES) and ex situ (NMR) techniques to follow the rearrangement process and to study the mechanism of VPT transformation involving intermediate structures. Diffraction, adsorption and microscopy data unambiguously showed that the transformation of **IWW** germanosilicate into the new zeolite phase (designated as IPC-18) proceeds without degradation of zeolite layers and thus affects only interlayer structure units without defects from structural rearrangement.

## Results

**3D-2D-3D zeolite transformation.** The unsuccessful application of the ADOR strategy for the 3D-2D-3D transformation of specific zeolites, including **IWW**[14], prompted us to develop alternative approaches for controllable disassembly-reassembly of appropriate zeolite frameworks containing labile structural units. The key step in such transformations is the hydrolysis of labile bonds (typically, Si–O–Ge or Ge–O–Ge) in double-four-ring (D4R) units connecting zeolite layers[15]. Severe hydrolysis conditions and inappropriate chemical composition of parent materials are among the most common reasons for the partial deterioration or even full collapse of zeolite frameworks during the disassembly[16]. Three types of germanosilicates with general formula $Si_xGe_{1-x}O_2$ and with chemical compositions ($Si/Ge = x/(1 − x)$) similar to those used for conventional ADOR transformations[16] were chosen to implement the VPT 3D-2D-3D rearrangement: **UTL** ($Si/Ge = 4.2$), **UOV** ($Si/Ge = 3.1$) and **IWW** ($Si/Ge = 3.7$). The structures of these initial germanosilicates are schematically shown in Fig. 1 as the combinations of dense (for **UTL**) or porous (**UOV**, **IWW**) layers and double-four-ring (D4R)

building units connecting the layers. Because **UTL** and **UOV** zeolites have been already transformed into respective daughter zeolites by conventional ADOR strategy[12,17], they were chosen in this study as reference materials to evaluate any differences between VPT and conventional ADOR approaches for known {parent zeolite–daughter zeolite} pairs. For the VPT transformation of the selected zeolites, a thin layer of zeolite powder was placed in the reactor containing a membrane permeable to vapours of low-molecular mass reactants located at the bottom part of the reactor (Fig. 1). The rearrangement of the zeolite was initiated by the interaction between acid vapour (12 M water solution of HCl with $p_{H2O} = 10$ mm Hg and $p_{HCl} = 23$ mm Hg at 303 K[18]) and the frameworks{–Si–O–Ge–} + HCl → {–Si–OH} + {Cl–Ge–}.

Volatile germanium chloride ($bp_{GeCl4} = 359$ K, for pure $GeCl_4$ $p_{GeCl4} = 94$ mm Hg at 300 K[19]) resulting from this interaction is adsorbed/dissolved and hydrolysed in the water solution ($GeCl_4 + nH_2O \rightarrow Ge(OH)_nCl_{4-n} + nHCl$), at the bottom of reactor. Such an approach allows unrestricted mass transport of the species formed by destruction of the most labile Ge-rich connecting units but avoids the deep reconstruction of the germanosilicate framework mediated by water.

Condensation of silanols ({–Si–OH} + {–Si–OH} → {Si–O–Si} + $H_2O$), formed during the hydrolysis, resulted in the formation of the respective daughter structures from parent zeolites (Fig. 1). The extent of the structural rearrangement differs among the zeolites used. Treatment of **UTL** leads to the formation of the known IPC-7 material (Supplementary Fig. 1) with alternating connectivity between layers through single-four-ring (S4R) and D4R units[17], whereas transformation of **UOV** results in the formation of the recently reported IPC-12 zeolite (Supplementary Fig. 1) with direct connectivity between layers, instead of the D4R units that are present in the initial framework[12]. The transformation of **IWW** results in the formation of a structure in which the layers are exclusively connected through S4R units (see discussion of this structure below). Therefore, the building units formed through VPT rearrangement depend on the structure of initial zeolite: the presence of pores in the layers of **UOV** and **IWW** zeolites results in facilitated mass transport and thus in the formation of S4R (for **IWW**) or O-bridge (for **UOV**) units, whereas treatment of **UTL** with non-porous layers produces material with alternating D4R/S4R units, under the same conditions.

IPC-7 (derivative of **UTL**) and IPC-12 (derivative of **UOV**) materials obtained using liquid-phase ADOR method don't have any major difference in properties with their analogues synthesised by VPT. In contrast, the open framework of layers in **IWW** zeolite cannot withstand the conventional ADOR treatment. Nevertheless, at room temperature, VPT yields an intermediate derivative (denoted as IPC-18P, Supplementary Fig. 2) that can be condensed at high temperatures to form the highly crystalline IPC-18 zeolite. The formation of S4R, but not D4R + S4R or O-bridge connections, between layers of IPC-18 can be explained as follows. A simulation study[20] predicted the **IWW** derivative with S4R connections to have lower formation enthalpy (by 3.5 kJ mol$^{-1}$) than the structure with O-bridges. Conversely, under the conditions applied, mass transport can be inhibited during the transformation. Low-molecular weight siliceous products of hydrolysis of D4R units can remain in the zeolite pores and subsequently participate in condensation to form S4R units, as observed in some ADORable zeolites, such as **OKO**[21]. As expected, the final IPC-18 material is significantly Ge-depleted (Si/Ge decreased from 3.7 to 65.0), thus indicating that germanium atoms are successfully removed from the **IWW** framework through the formation of volatile species (presumably $GeCl_4$).

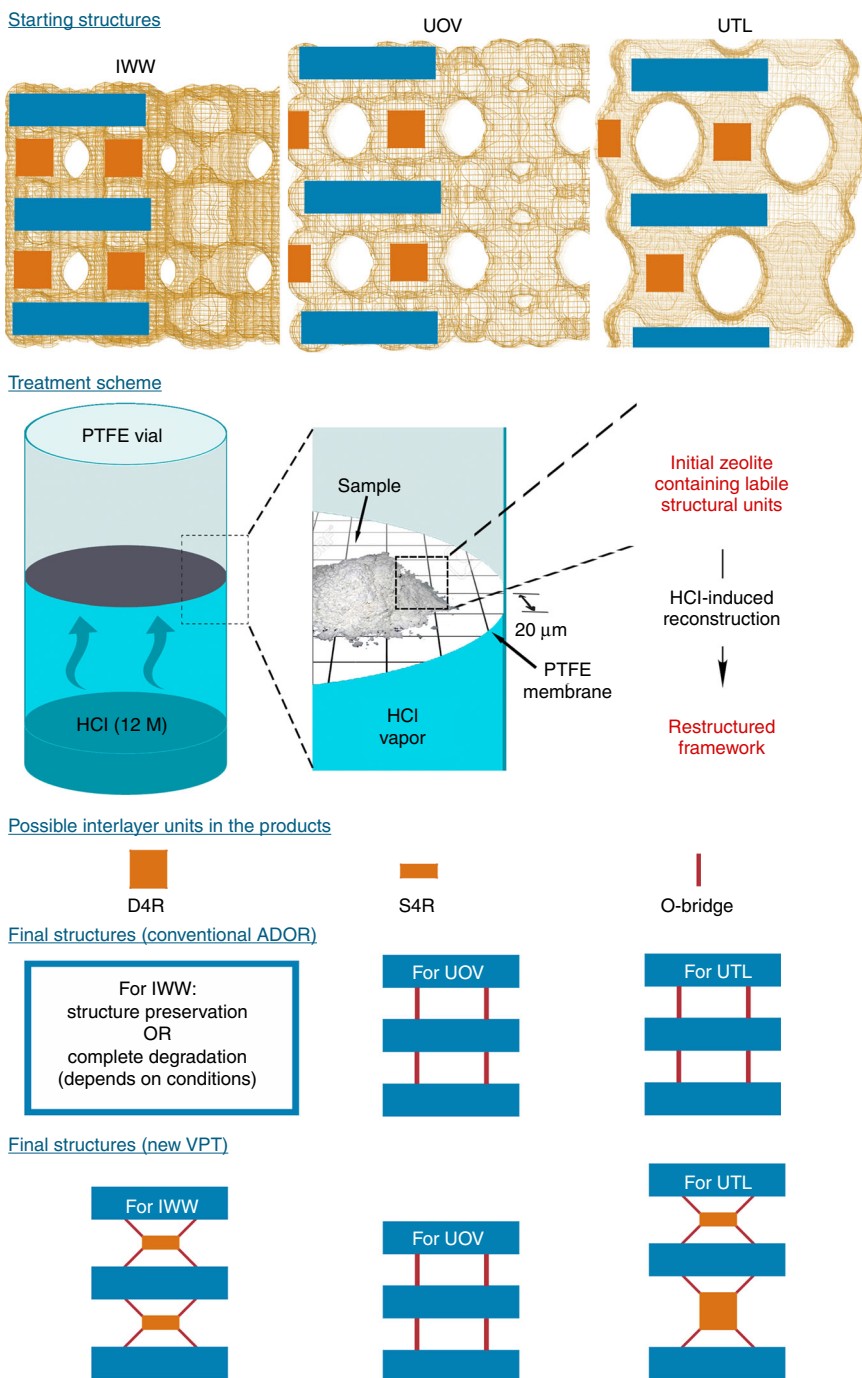

**Fig. 1** Starting materials for VPT and respective products. Structures of initial zeolites are porous frameworks composed of layers (blue tetragons) and connecting D4R units (orange squares). Vapour of HCl (blue on treatment scheme of the VPT method) interacts with solid zeolite resulting in structural transformation accompanied by leaching of Ge atoms probably as GeCl₄. Respective D4R units can be either reconstructed to S4Rs or decomposed. Final structures obtained via conventional ADOR and VPT rearrangement of the parent zeolites (**IWW**, **UOV** and **UTL**) are different due to the difference in layers porosity, thus in the interlayer species diffusion rate

**Structural features of IPC-18 vs parent IWW**. The XRD pattern of IPC-18 matches well that of the theoretically predicted framework (Fig. 2a). Based on Rietveld refinement of the final structure (Fig. 2b, c), the space group was identified as $P2_1/c$, which differed from the parent **IWW** (*Pbam*). The following parameters of the cell were determined: $a = 9.606(4)$ Å, $b = 12.7280(21)$ Å, $c = 40.717(7)$ Å, $\alpha = 90.0°$, $\beta = 94.97°$, $\gamma = 90.0°$. The parent **IWW** zeolite has a 3D pore system with 8- and 12-ring channels, which are both intersected by sinusoidal 10-ring channels. After VPT rearrangement, 8- and 12-ring pores located along the 001 projection remain intact, whereas 10-ring pores become eight-ring pores due to the transformation of D4Rs into S4Rs. This transformation of the pore system resulted in decreased micropore volume and average pore size (Fig. 2d, e). The micropore volume of IPC-18 is 1.7 times lower than that of the initial **IWW** sample (0.172 vs. 0.104 cm³ g⁻¹, respectively). The average channel diameter decreases from 0.63 nm, for **IWW**, to 0.58 nm, for IPC-18. Despite differing in micropore volume and size, both parent and daughter zeolites show the same {mesopore + interparticle} volume (0.025 cm³ g⁻¹) and similar

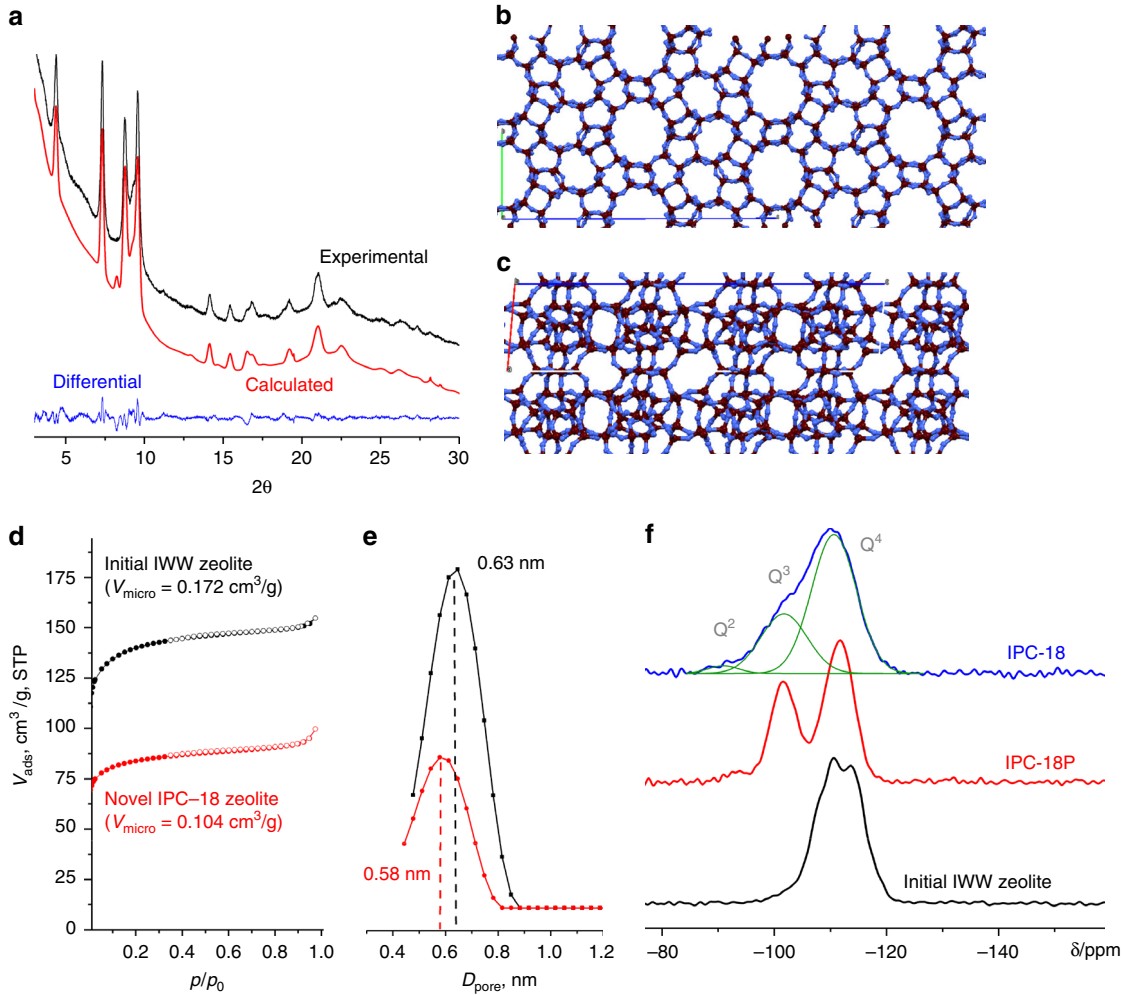

**Fig. 2** Basic characterisation of IPC-18. **a** XRD patterns of IPC-18: experimental (black), calculated after Rietveld refinement (red), and the difference between them (blue); **b**, **c** crystallographic models of IPC-18 in the *ab* projection demonstrating 12 and 8-ring pores (**b**), and in *ac* projection, showing connectivity between layers through S4R units (**c**); **d** nitrogen adsorption and desorption isotherms for parent **IWW** and daughter IPC-18 zeolites; **e** pore size distribution for initial **IWW** (black) and final IPC-18 (red); **f** $^{29}$Si MAS NMR spectra of **IWW**, intermediate IPC-18P and IPC-18 samples

morphology of the crystals, ~1 μm in size (Supplementary Fig. 3). This result indicates that no significant fraction of additional pores, which would influence the textural properties of IPC-18, is created during the VPT transformation. This is an important conclusion because the state of framework silicon atoms is significantly changed in the course of the structural transformation.

Solid-state $^{29}$Si MAS NMR analysis of **IWW**, intermediate IPC-18P and IPC-18 shows the evolution of the degree of condensation of silanols in this process (Fig. 2f). The spectrum of the starting **IWW** is characterised by the presence of a signal at −115 ppm, which corresponds to $Q^4$ Si atoms. VPT treatment results in the hydrolysis of most of Si–O–Ge bonds and formation of the IPC-18P precursor, which increases the intensity of silanol groups ($Q^3$ atoms, −101 ppm) in the spectrum. The spectrum of IPC-18P also contains a low-intensity peak at −93 ppm, which can be related to $Q^2$ silicon atoms. This is an unusual feature of the zeolitic intermediate obtained by disassembly because a recent in situ NMR study revealed only $Q^3$ Si among deficient atoms during a top-down structural transformation of germanosilicate[15]. Subsequent thermal treatment of IPC-18P decreases the intensity of the signal attributed to silanol groups due to topotactic condensation of the zeolite layers into a 3D structure, with no evidence of additional defects in IPC-18 due to the reconstruction of the pore system in **IWW**. However, the NMR

spectrum of the final IPC-18 still contains quite intense (25–30 % of overall area fraction) $Q^3$ signal (Fig. 2f). IPC-18 synthesis (IPC-18P reassembly) temperature lower than optimal can be the reason for relatively high content of silanols in final material. In addition, some Ge atoms are still remained in IPC-18 framework as $Si(OGe)(OSi)_3$ domains that give a peak, which can overlap with the $Q^3$ peak.

Structural differences and similarities between the parent **IWW** and its derivative can be directly observed through spherical aberration corrected ($C_s$-corrected) Scanning Transmission Electron Microscopy (STEM) analysis of the respective materials (Fig. 3). Images collected from the top view of the layers (Fig. 3a, c) along the [100] orientation show the same member rings and the same pore arrangements, thus confirming their preservation during the VPT treatment, as well as topotactic condensation. Along this orientation, the D4Rs that are affected by the hydrolytic process are not visible in these top views. Therefore, after collecting data from the side view along [010], the interlayer distances significantly decreased due to changes in D4Rs which have now turned into S4Rs. The projected measured distance between "T" atoms for a D4R is ~3.1 Å. Thus, initially, a decrease of ~3 Å should be expected. The distances between layers directly measured on high-resolution images are 12.23 Å for **IWW** and 9.11 Å for IPC-18, in very good agreement with the

**Fig. 3** Comparison of different projections of IWW and IPC-18. $C_s$-corrected STEM-ADF images of **IWW** (**a**, **b**) and IPC-18 (**c**, **d**) zeolites: **a**, **c**—top [100] view of the layers, which remain intact during the VPT rearrangement (crystallographic models of the same layer are superimposed for clarity), **b**, **d**—side [010] view showing the decrease in interlayer distance caused by the transformation of D4Rs into S4Rs

transformation of the linking units. Selected area electron diffraction (SAED) and/or Fourier diffractograms (FD) measurements were also collected along different orientations (see Fig. 3 insets), which clearly showed, first, the good crystallinity of the materials, with no additional structural defects, and, second, the formation of S4Rs. The analysis of HRTEM images collected in three projections of **IWW** crystals (Supplementary Fig. 4a–c) and IPC-18 (Supplementary Fig. 4d–f) confirms the aforementioned conclusion about the direction in which the anisotropic structural transformation occurs. For one projection [001], d-spacing is the same (12.0 Å) for both parent and daughter zeolites. The major difference between **IWW** and IPC-18 is the connectivity between layers (Supplementary Fig. 4).

**Mechanism of VPT zeolite rearrangement**. Characterisation of the starting and final zeolite structures is relatively simple because they are crystalline materials with well-ordered frameworks. In contrast to the relatively stable IPC-18P layered material, any intermediate structures formed during the **IWW**-to-IPC-18P transformation (hydrolysis) or during the IPC-18P-to-IPC-18 condensation (reassembly) are hardly detectable without in situ methods as both processes are completed within 1 h. We used in situ XRD to trace the diffraction peaks with different Miller indices $hkl$-related to different directions in the structures of the treated materials ($hkl$ ($l \neq 0$) interlayer peaks, with varying positions over time, and intralayer $hk0$ reflections, which should maintain the positions) to shed the light on the whole mechanism of **IWW**-to-IPC-18 transformations (a general scheme is presented in Fig. 4e). In addition, we performed quantitative lattice analysis based on XRD patterns and followed the local state of the Ge atoms by in situ XANES during the VPT rearrangement to identify the relationship between the nature of leaving atoms and framework features at each stage of the structural transformation.

Immediately after starting the treatment (<1 min), the positions of $00l$ reflections shifted to higher angles (Fig. 4a), indicating a decrease in d-spacing caused by disassembly of D4R connecting units. Surprisingly, the positions and intensities of $hk0$ reflections (110 and 400 are the most representative, Fig. 4a) are also significantly changed already at the first step of hydrolysis (0–2 min under conditions used). Analysis of the correlation between unit cell parameters and treatment time provided further insight into the processes that occur during the first step of the VPT (Fig. 4c). Not only the interlayer $c$ parameter but all lattice constants ($a$, $b$, $c$) considerably change during this step, indicating partial distortion of the structure of **IWW** layers (illustrated on Fig. 4e). Simultaneously, the $\beta$ angle, which is indicative of the shift of layers relative to each other, remains unchanged in the first step. Thus, at this stage, the layers become flexible but are still stacked, most likely due to ionic interactions and hydrogen bonding between interlayer species—primary products of hydrolysis. In situ XANES (Fig. 4d) revealed the major change in the nature of Ge atoms during this period since the Ge K-edge ($E_0$) steeply and linearly decreased over time at the same stage (0–2

min) because most Ge–OT$^{IV}$ (T$^{IV}$ = Si or Ge) bonds predominantly located at D4R units were broken. The intensity of some XRD reflections (related to both inter- and intralayer planes, e.g., 110 and 111) decrease in the first step of the VPT transformation, suggesting a complete loss of structural ordering in the respective directions of the lattice.

During the second step of hydrolysis (from 2 to 10–12 min), the aforementioned reflections appear at slightly shifted positions, indicating at least a partial recovery of the respective framework ordering. At first sight, the positions of the other $hkl$ ($l \neq 0$) reflections continue the gradual right-shift in the second stage of the rearrangement (Fig. 4a). However, the behaviour of the respective lattice constant ($c$) is more complex (Fig. 4c): after a series of minor changes (at ~½, 2, 3 min when $c$ parameter is either slightly decreased or increased), $c$ starts to decrease, with a clear exponential decay trend (Fig. 4c, blue line) until the end of the procedure. The parameters $a$ and $b$ monotonically decrease during the second step of hydrolysis, whereas $\beta$ peaks around 90.7°, indicating a slight, transient shift of the layers relative to each other during the structural reorganisation of the layers. These processes are accompanied by a linear change in $E_0$ as a function of time in XANES spectra, which is slower than the first step (Fig. 4d). Additional peaks around $E = 11120$ and 11125 eV appear in XANES spectra at this stage (Supplementary Fig. 7), indicating an accumulation of hydrolysed Ge species extracted from the framework. Overall transformations during this period can be attributed to a slow hydrolysis of isolated Ge–O bonds remaining after primary decomposition of the D4Rs (major fraction) and extraction of Ge atoms located in zeolite layers (minor fraction because the layers are almost purely siliceous). Completion of the D4R disassembly and removal of intercalated species allows the partial recovery of layer organization as the parameters $b$ and $\beta$ return to the original values after 12 mins of treatment.

At the final, third, step of the VPT rearrangement (12 + min), most XRD reflections maintain their positions and intensities, and only a few $hkl$ ($l \neq 0$) reflections, such as 001 and 313, show changes in their intensity after 12 min (Fig. 4a), similarly to the lattice parameters (Fig. 4c) and to the Ge state, which remains virtually unchanged during the final stage of hydrolysis (Fig. 4d).

Heating the intermediate IPC-18P material up to ~500 K has no effect on the positions or intensities of intralayer $hk0$ reflections, highlighting a complete layer rearrangement during the hydrolysis step of the VPT transformation (Fig. 4b). In contrast, the position of the 001 reflection in relation to the interlayer distance gradually shifts in the range of $T = $ RT $- 350$ K, being almost constant at temperatures of 350–450 K. Further increasing the temperature results in a gradual collapse of the IPC-18 zeolite framework, accompanied by a shift in all XRD reflections at $T = 450$–650 K and in a decrease of their intensities at T >650 K (Fig. 4b). Based on our in situ XRD and XANES data, we propose the following mechanism of **IWW**-to-IPC-18 transformation, as illustrated in Fig. 4e:

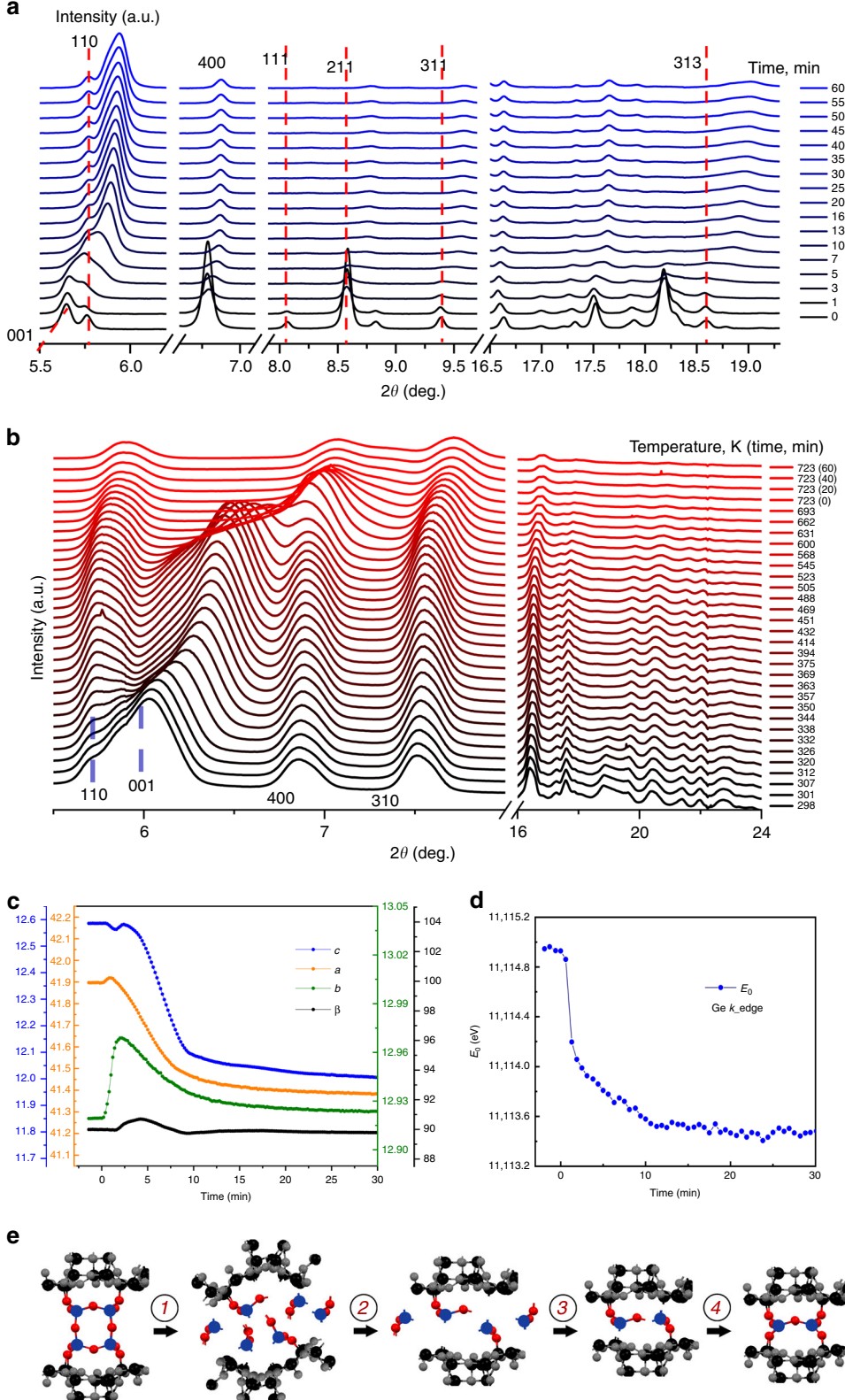

(1) fast hydrolysis of Ge-rich linking units and partial distortion of the silicon-enriched layers at RT (0–2 min under the conditions used in this study);

(2) intermediate extraction of the remaining Ge atoms accompanied by partial reconstruction of the initial framework, particularly of the layers (2–12 min);

(3) slow rearrangement of the species remaining in the interlayer space until reaching the equilibrium state at RT (>10–12 min)

(4) layers condensation (reassembly) at slightly increased temperatures (up to 350 K) with no effect on the structure of the layers.

**Fig. 4** In situ data supporting the assumed mechanism of VPT. **a** In situ XRD demonstrating changes during the hydrolysis of the parent **IWW** zeolite; **b** in situ XRD collected during thermal treatment of IPC-18P; **c** variation of lattice parameters (lengths in Å and angle in degree) as a function of time during the VPT treatment; **d** evolution of the primary line extracted from in situ XANES spectra representing changes in Ge state during hydrolysis steps; **e** proposed mechanism (Si and Ge atoms are not distinguished for clarity and only structural, but not chemical, changes are shown). Full in situ XRD data, lattice parameters and XANES profiles are shown in Supplementary Figs. 5, 6 and 7, respectively. Analysis of the variation of lattice parameters and position of primary line in XANES spectra allows to distinguish several steps during VPT. Unexpectedly, the disassembly of Ge-rich D4Rs was accompanied by partial disorder of the layers, which become more crystalline on the second step of hydrolysis. The topology of resulting layers only slightly differs from that of starting zeolite. Following deintercalation of interlayer debris and condensation allows the formation of new IPC-18 zeolite

The VPT rearrangement technique was developed and first exemplified for the ADOR transformation of different germanosilicate zeolites characterised by the unidirectional location of Ge-enriched D4R units (i.e., **UTL**, **UOV**, **IWW**). The VPT approach overcomes the limitations of the classical ADOR approach for the rational design of zeolites. Although VPT treatment of **UTL** and **UOV** germanosilicates produces known ADORable IPC-7 and IPC-12 zeolites of high crystallinity, VPT rearrangement of the **IWW** framework enabled preparation and structure refinement of the previously predicted, yet thus far inaccessible, IPC-18 zeolite via existing synthesis approaches. In situ XRD and XANES studies on the VPT rearrangement of **IWW** revealed the acid-induced reorganisation of not only interlayer but also intralayer framework domains, resulting in the temporary disordering of the structure followed by its reconstruction to a well-ordered material with a different framework from the parent zeolite. The successful application of VPT rearrangement to germanosilicate zeolites of different topologies highlights the potential of this technique for the 3D-3D transformation of crystalline materials with labile frameworks collapsing upon contact with the solvent (e.g., ordered organic-free germanates or some metal-organic frameworks). In combination with other methods for post-synthesis alteration of 3D frameworks, this approach makes it possible to manipulate the structures of anisotropically labile materials.

## Methods

**Synthesis of MPPH.** The structure-directing agent (SDA) for IWW synthesis is 1,5-Bis-(methylpyrrolidinium)pentane dihydroxyde (MPPH), which was prepared according to[22]. In total, 40 g of N-methylpyrrolidine (Sigma Aldrich, 97%) was mixed with 37.5 g of 1,5-dibromopentane (Sigma Aldrich, 97%) in 300 ml of acetone (Lachner, 99.99%) and heated under reflux for 24 h. The resulting 1,5-bis-(methylpyrrolidinium)pentane dibromide was then ion-exchanged into th ehydr-oxide form using Ambersep® 900(OH) an anion exchange resin (Acros Organics, 0.8 mmol of SDA per 1 g of anion exchange resin). The solution was concentrated under low pressure (35 Torr) at 303 K until the hydroxide concentration was ~1.0 M.

**Synthesis of DMAH.** (6R,10S)-6,10-dimethyl-5-azoniaspiro[4,5] decane hydroxide (DMAH), the SDA for the synthesis of germanosilicate **UTL**, was prepared based on the procedure reported in Ref. [23]. In total, 16.07 g of (2R,6S)-2,6-dimethylpiperidine (Sigma Aldrich, 98%) was added drop-wise to 140 ml of water solution containing 5.68 g of sodium hydroxide (Penta, 98%) and 30.66 g of 1,4-dibromobutane (Aldrich, 99%). Subsequently, the mixture was refluxed under intensive stirring for 12 h. After cooling in an ice bath, an ice-cooled 50% (wt.) solution of NaOH (70 ml) was added, and further solid NaOH was added until forming the oil product. After crystallisation, the solid was filtered and extracted with chloroform (Lachner, 99.92%). The organic fraction was dried, using anhydrous sodium sulphate (Sigma Aldrich, 99%), and partly evaporated (50–100 mL of residual volume); then, diethyl ether (Lachner, 99.97%) was added to the remaining mixture. The final solid product was washed three times with diethyl ether. The bromide salt was ion-exchanged into the hydroxide form using an Ambersep® 900 (OH) anion exchange resin (0.8 mmol of SDA per 1 g of anion exchange resin) and concentrated by evaporation to prepare a 1 M solution.

**Synthesis of IWW zeolite.** IWW zeolite samples was performed according to Ref. [22] from the reaction mixture 0.66 $SiO_2$: 0.33 $GeO_2$: 0.25 MPPH: 15 $H_2O$. Appropriate amounts of germanium dioxide (Sigma Aldrich, 99.99%) and tetraethyl orthosilicate (Sigma Aldrich, 98%) were added to the SDA solution under stirring. The resulting mixtures were stirred to evaporate the ethanol formed by hydrolysis of tetraethyl orthosilicate. Then, the gels were heated in Teflon-lined stainless steel autoclaves at 448 K for 11 days. The final products were recovered by

centrifugation, washed with water and dried at 333 K overnight. The resulting solids were calcined at 853 K for 6 h in air.

**Synthesis of UTL zeolite.** UTL zeolite samples was based on the method reported in Ref. [23] by crystallisation of a gel with the composition of 0.66 $SiO_2$: 0.33 $GeO_2$: 0.25 DMAD: 30 $H_2O$, at 448 K for 6 days under agitation (60 rpm). The solid products were separated by filtration, washed out with distilled water, and dried overnight at 368 K. The final solids were calcined at 823 K for 6 h with a temperature ramp of 2 K min$^{-1}$ under air flow (200 ml min$^{-1}$).

**Synthesis of UOV zeolite.** Samples of germanosilicate UOV were prepared according to Ref. [24] from reaction mixtures with the following composition: 0.33 $SiO_2$: 0.66 $GeO_2$: 0.25 DMDH: 10 $H_2O$, using decamethonium dihydroxide (DMDH) as the SDA. DMDH was previously prepared from the bromide form by ion exchange using Ambersep® 900(OH) anion exchange resin. The solution of DMDH was concentrated under low pressure (25 Torr) at 303 K until an SDA concentration >1.5 M. Some germanium oxide was dissolved in a mixture of water and DMDH. Silica (Cab-O-Sil M5, Supelco Analytical) was gradually added to the solution, and the mixture was stirred at room temperature for 30 min. The reaction gels were autoclaved at 448 K for 7 days under static conditions. The solid product was recovered by centrifugation, washed several times in distilled water until the pH of the solution became neutral, dried at 338 K during 12 h, and finally calcined at 823 K for 6 h with a temperature ramp of 2 K min$^{-1}$ under air flow (200 ml min$^{-1}$).

**VPT treatment.** In total, 0.1 g of calcined zeolite was placed in the reactor containing polytetrafluoroethylene (PTFE) membrane over 10 ml of 12 M hydrochloric acid solution (p. a., Penta) at 298 K for $\tau = 5$ min–48 h. Caution: the VPT procedure uses concentrated HCl. Standard laboratory practice for dealing with highly corrosive liquids (including appropriate personal protective equipment) should be used.

**Basic characterisation.** The structure and crystallinity of zeolites were determined by powder X-ray diffraction using a Bruker AXS D8 Advance diffractometer with Cu Kα radiation in Bragg-Brentano geometry. The chemical composition was determined on an ICP/OES (ThermoScientific iCAP 7000). Adsorption experiments were performed using nitrogen as a probe molecule. Ad-/desorption isotherms were measured using an ASAP 2020 (Micrometeritics) static volumetric apparatus at liquid nitrogen temperature (−196 °C). Before the sorption measurements, all samples were degassed with a turbomolecular pump at 523 K for 4 h. The micropore volume was estimated by application of the t-plot method, and the pore size distribution was obtained using the NLDFT method. The morphology of the samples was determined under a scanning electron microscope (SEM) TES-CAN Vega. Solid-state $^{29}$Si NMR spectra were recorded on an Agilent DD2 500WB spectrometer at a resonance frequency of 99.30 MHz. MAS NMR measurements were carried performed with a commercial 3.2 mm triple resonance MAS probe. The chemical shifts of $^{29}$Si are referenced to tetramethylsilane at 0 ppm.

**Synchrotron XRD.** The in situ synchrotron XRD was performed at the BL14B and BL19U2 stations of the Shanghai Synchrotron Radiation Facility (SSRF). The wavelength of incident X-ray photons was 0.1240 nm, and the diffraction patterns were recorded using two-dimensional X-ray detectors. To monitor the hydrolysis process in real-time, the sample was sealed in a homemade cell with Kapton windows. Diffraction patterns from the sample were continuously recorded with an exposure time of 10 s per frame and with an interval of 2 s between adjacent frames. The beginning of reaction was controlled by an electric pump, which injected concentrated hydrochloric acid into the cell to initiate the hydrolysis. For the in situ heating experiment, the sample was placed on a heating stage. The temperature was increased from room temperature to 723 K at a rate of 1.5 K per minute and was then kept at 723 K for 1 h. Diffraction patterns were continuously recorded at an exposure time of 100 s per frame and at an interval of 20 s between adjacent frames.

**Quantitatively analysis for the lattice parameters.** For the XRD patterns obtained during the in situ hydrolysis step of the VPT transformation, the time-dependent peak positions of reflections (001), (320), (400), (120), (311), (310)

and (211), as shown in Fig. SI-6, are used to calculate the quantitative lattice information with the fsolve function in Matlab, assuming a monoclinic crystalline structure for the materials.

**Synchrotron XANES**. Ge K edge XANES measurements were performed at the beamline 1W1B of the Beijing Synchrotron Radiation Facility (BSRF). The spectra were collected in the transmission Q-XAFS mode to monitor the hydrolysis reaction, with a rate of 37.6 s per spectrum. During the experiment, the sample was sealed in the homemade cell, as described above.

**Electron microscopy**. Electron microscopy analyses were performed in a coldFEG JEOL Grand ARM 300 electron microscope. The microscope was equipped with a double corrector from JEOL assuring a spatial resolution of 0.7 Å when operated at 300 kV. Before observation, the samples were deeply crushed using a mortar and pestle and dispersed in ethanol. A few drops of the suspension were deposited on a holey carbon copper microgrids. For imaging, both Annular Dark Field and Annular Bright Field detectors were used simultaneously, acquiring data in both detectors.

## Data availability

XRD, NMR, adsorption and in situ XRD data were deposited at 10.6084/m9.figshare.9853502 and 10.6084/m9.figshare.9853505. All the data that support the findings of this study are available from the corresponding authors upon reasonable request

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

## Acknowledgements

M.S., M.M., R.E.M., J.Č. and M.O. acknowledge OP VVV "Excellent Research Teams" project No.CZ.02.1.01/0.0/0.0/15_003/0000417– CUCAM. M.S. and M.O. thank the Primus Research Program of the Charles University (project number PRIMUS/17/SCI/22 "Soluble zeolites"). R.E.M. also thanks the ERC (Advanced Grant 787073 "ADOR"). A.M. acknowledges The Centre for High-resolution Electron Microscopy (ChEM), supported by SPST of ShanghaiTech University under contract No. EM02161943, and the Natural National Science Foundation of China, through projects NFSC-21850410448 and NSFC-21835002. Z.L. acknowledges the support from the National Key Research and Development Program of China (2016YFA0300102) and the National Natural Science Foundation of China (11675179, 11434009). J.Č. acknowledges the support of the Czech Science Foundation to the project EXPRO (19-27551×).

## Author contributions

V.K. and M.S. performed synthesis study and wrote respective sections of the article, Q.Y. performed lab-scale optimisation of the synthesis and basic characterisation of materials, A.M. and M.M. were responsible for microscopy and collecting of structural data, J.Z. and Z.L. designed and performed in situ experiments and respective analysis, R.E.M. and J.Č. analysed the results and supervised particular directions of the study, M.O. designed the synthesis strategy, analysed the results and wrote the article.

## Competing interests

The authors declare no competing interests.
