## [Peer Review File · Nature Communications]

Review of NCOMMS-19-23277-T

This manuscript presents a new type of ADOR (assembly-disassembly-organisation-reassembly) process, which was first established by the coauthors. Although there have been a number of papers published on ADOR processes, there are two unique aspects of this work that I feel warrant its publication in Nature Communications. The first is the ability to use a new zeolite framework, IWW, as the parent phase for structural transformation into a new (daughter) structure, ICP-18. The second point, and arguably the most impressive, is the use of a new technique to accomplish the IWW to ICP-18 transformation, which is a vapor-phase-transport strategy. As described in Figure 1, the use of this new technique overcomes obstacles in previous studies that have failed to convert IWW into other structures. The authors demonstrate that similar strategies in the liquid phase have been unsuccessful. Therefore, the identification of a new structure coupled with the unique method to achieve the transformation that is demonstrated in this manuscript are unique and impactful in the field of microporous materials synthesis and design. The paper is well written and the authors use a wide range of impressive techniques to confirm their conclusions. Overall, this is a very comprehensive study and there are few scientific weaknesses that can be identified. There are several minor comments that the authors should address, which are listed below. Provided that they do so, it is the opinion of this reviewer that the manuscript should be accepted for publication.

Minor Comments:

- Regarding the NMR data in Figure 2f, the authors claim that there is only a minor fraction (<5%) of Q³ species in ICP-18; however, based on the visual appearance of the spectrum, this fraction seems to be much larger. The authors should double check their calculations.
- In Figure 3, images in (b) and (d) should be the same scale to better facilitate the comparison of interlayer distances.
- The x-axis label of Figure 4a is cut off.
- In Figure 4c, should there be units for the lattice parameters (none are listed in the figure or caption)?

Reviewer #2 (Remarks to the Author):

While the subject of this manuscript is interesting, the manuscript itself is very poorly written. I provide a few comments. The three letter codes should be defined. Previous use of VPT should be provided with a few references. The results and discussion use Fig. 1. This figure is not well done. Figures 1a and 1c are very difficult to understand if one does not know a lot about molecular sieves. It would be helpful to show the IPC-18 structure early in the manuscript. I don't like the word novel. What is novel about this structure. It is just a new structure. Also, in this opening section, it would be good to show some chemical formulas to illustrate what is occurring at the molecular level. Since the authors claim is that GeCl_4 is removed and in the liquid phase at the end of the reaction -- prove it. There is no mass balance shown for any reaction. Does the Ge content in the liquid phase make sense with the proposed process? Are the samples prepared from UTL and UOV by the VPT the same as by ADOR? If so, great, if not, why? What about safety issues? None are listed. Characterization data in Figs. 2 and 3 look good.

Dear Editor and Referees,

We are grateful for your useful comments to our manuscript. Here we address point-to-point all questions and concerns. Also, as two reviewers have opposing opinions about the style of manuscript (“paper is well written” and “manuscript itself is very poorly written”), we have tried to improve it making introduction and discussion more “friendly” for general readers.

Reviewer #1 (Remarks to the Author):

This manuscript presents a new type of ADOR (assembly-disassembly-organisation-reassembly) process, which was first established by the coauthors. Although there have been a number of papers published on ADOR processes, there are two unique aspects of this work that I feel warrant its publication in Nature Communications. The first is the ability to use a new zeolite framework, IWW, as the parent phase for structural transformation into a new (daughter) structure, IPC-18. The second point, and arguably the most impressive, is the use of a new technique to accomplish the IWW to IPC-18 transformation, which is a vapor-phase-transport strategy. As described in Figure 1, the use of this new technique overcomes obstacles in previous studies that have failed to convert IWW into other structures. The authors demonstrate that similar strategies in the liquid phase have been unsuccessful. Therefore, the identification of a new structure coupled with the unique method to achieve the transformation that is demonstrated in this manuscript are unique and impactful in the field of microporous materials synthesis and design. The paper is well written and the authors use a wide range of impressive techniques to confirm that the conclusions. Overall, this is a very comprehensive study and there are few scientific weaknesses that can be identified. There are several minor comments that the authors should address, which are listed below. Provided that they do so, it is the opinion of this reviewer that the manuscript should be accepted for publication.

Minor Comments:

- Regarding the NMR data in Figure 2f, the authors claim that there is only a minor fraction (<5%) of Q3 species in IPC-18; however, based on the visual appearance of the spectrum, this fraction seems to be much larger. The authors should double check their calculations.

We performed deconvolution of the mentioned NMR spectrum of IPC-18. According to the obtained data, the fraction of Q3 in IPC-18 is 25 – 30 % that confirms Referee’s suggestion. Respective part of the manuscript was modified.

- In Figure 3, images in (b) and (d) should be the same scale to better facilitate the comparison of interlayer distances.

Scale on image (b) was increased to make it similar to (d) as it was suggested by Referee.

- The x-axis label of Figure 4a is cut off.

The graph was modified to exclude the displacement.

- In Figure 4c, should there be units for the lattice parameters (none are listed in the figure or caption)?

Respective information was added to the figure caption.

Reviewer #2 (Remarks to the Author):

While the subject of this manuscript is interesting, the manuscript itself is very poorly written. I provide a few comments.

- The three letter codes should be defined.

Definition for three letter codes was added to the main text at first mention (“three letters codes are assigned to established structures of zeolites that satisfy the rules of the IZA Structure Commission”)

- Previous use of VPT should be provided with a few references.

Proposed VPT approach for ADOR transformation of zeolite is not based on any vapour-related methods used in zeolite chemistry up to now. Existing techniques that use vapour phase conversion e.g. for developing of mesoporosity or modifying of chemical composition) have no relation to the discussed VPT rearrangement besides the fact that some of the processes proceed in gas phase or are assisted by vapour.

- The results and discussion use Fig. 1. This figure is not well done. Figures 1a and 1c are very difficult to understand if one does not know a lot about molecular sieves. It would be helpful to show the IPC-18 structure early in the manuscript.

Figure 1 was completely reconstructed to represent the starting materials, procedure details and final products in more logic and clear way.

- I don't like the word novel. What is novel about this structure. It is just a new structure.

Word “novel” has been replaced with more appropriate “new” in the text of the manuscript. Of course, the dictionary definition of the adjective novel is actually ‘new’ (sometimes ‘interestingly new’) so the distinction between the words new and novel is really not all that well established. However, novel is sometimes seen as a bit clichéd in scientific writing (which we think is what the reviewer is getting at) – and with that we think we can agree and so we are very happy to replace novel with new.

- Also, in this opening section, it would be good to show some chemical formulas to illustrate what is occurring at the molecular level.

Respective modifications were made at the beginning of discussion part:

p.2: The key step in such transformations is the hydrolysis of labile bonds (typically, Si–O–Ge or Ge–O–Ge)...

p.2: Three types of germanosilicates with general formula $\text{Si}_x\text{Ge}_{1-x}\text{O}_2$ and with chemical compositions ($\text{Si}/\text{Ge} = x/(1-x)$) similar...

p.3: interaction between acid vapor ... and the frameworks $\{-\text{Si}-\text{O}-\text{Ge}-\} + \text{HCl} \rightarrow \{-\text{Si}-\text{OH}\} + \{\text{Cl}-\text{Ge}-\}$.

p.3: ... hydrolyzed in the water solution ($\text{GeCl}_4 + n\text{H}_2\text{O} \rightarrow \text{Ge}(\text{OH})_n\text{Cl}_{4-n} + n\text{HCl}$)...

p.3: Condensation of silanols ($\{-\text{Si}-\text{OH}\} + \{-\text{Si}-\text{OH}\} \rightarrow \{\text{Si}-\text{O}-\text{Si}\} + \text{H}_2\text{O}$)...

- Since the authors claim is that GeCl4 is removed and in the liquid phase at the end of the reaction -- prove it. There is no mass balance shown for any reaction. Does the Ge content in the liquid phase make sense with the proposed process?

Indeed, exact value of Ge content in the liquid phase (considered as a medium for byproducts) is not essential. Also, the particular state of Ge (partially GeOHxCl_y or fully $\text{GeO}_2\text{xH}_2\text{O}$ hydrolyzed products and thus the exact mass balance) was not under our investigation as it does not influence the output of the main process, i.e. zeolite transformation. Decrease of Ge content in a solid phase (from 21 to 1.5 mol %, discussed in the text as $\text{Si}/\text{Ge}=3.7$ and 65, respectively) is the proof for its removal from material under study. Thus, no changes were made in the text of the manuscript.

- Are the samples prepared from UTL and UOV by the VPT the same as by ADOR? If so, great, if not, why?

Discussion of the differences between outputs of ADOR and VPT for UTL and UOV is modified (p.4). Fig. 1 is also modified to show these differences clearly.

- What about safety issues? None are listed. Characterization data in Figs. 2 and 3 look good.

Safety issues associated with using concentrated acid solutions are added to the methods section.